# Application of Deep Supervised Learning to Nailfold videocapillaroscopy and Red Blood Cells Motion Detection

**Alexey Kornaev**[*][1]                                   AVKORNAEV@GMAIL.COM
**Dmitry Stavtsev**[1]                              STAVTSEV.DMITRY@GMAIL.COM
**Elena Kornaeva**[1]                                 SMKORNAEVA@GMAIL.COM
[1] *Orel State University named after I.S. Turgenev, Komsomolskaya St. 95, 302026 Orel, Russia*

**Mikhail Volkov**[2]                                      FOSP@GRV.IFMO.RU
[2] *ITMO University, Kronverksky Ave. 49, 197101 Saint Petersburg, Russia*

**Editors:** Under Review for MIDL 2021

## Abstract

The paper deals with processing data obtained using nailfold high-speed videocapillaroscopy. To detect the red blood cells speed two approaches are used. The deterministic approach is based on pixel intensities analysis for object detection and calculation of the red blood cells displacement and velocity in a vessel. The obtained data formulate targets for machine learning. The stochastic approach is based on a sequence of artificial neural networks. The semantic segmentation network UNet is used for vessel detection. Then, the classification network GoogLeNet or ResNet18 is used as a feature extractor to convert masked video frames to a sequence of feature vectors. And finally, the long short-term memory network is used to approximate the red blood cells velocity. The results demonstrated that the accuracy of the mean velocity approximation in the time range of several seconds is up to 0.96. But the accuracy at each specific time moment is less accurate. So, the proposed algorithm allows the determination of the RBCs mean velocity but it doesn't allow determination of the RBCs pulsations accurate enough.

**Keywords:** Videocapillaroscopy, Red blood cells, Semantic segmentation, Deep convolutional networks, Long short-term memory networks.

## 1. Introduction

The red blood cells (RBC) mechanical properties may indicate a healthy or diseased state of an organism. Modern equipment for video recording and processing allows the RBCs motion and deformation observation in vessels in real-time mode. Obviously, real-time in-vivo health condition monitoring is a promising method. This paper deals the RBCs velocity approximation using the long short-term memory (LSTM) networks. The study is based on results of nailfold videocapillaroscopy (VCS) presented in the paper (Stavtsev et al., 2019).

---

[*] Corresponding author

## 2. Data collection

Images of the nailfold capillaries were obtained using a custom videocapillaroscopy setup providing direct visualization of capillary blood flow. In this setup, a long working distance objective (Mitutoyo M Plan APO 5X) with 5×magnification and 0.14 aperture was used together with a Bi-Convex lens with 200 mm focal length. A green LED light source was mounted to the side of the objective to illuminate the area of nailfold bed at a distal phalanx of a finger. The use of green light is also associated with better light absorption by the blood in this part of the spectrum, which provides a better contrast. The video was recorded with CMOS camera (IDS UI-3060-C-HQ) at a resolution of 800×800 and 150 fps which allows the placing of several adjacent capillary loops in the field of view  Figure 1.

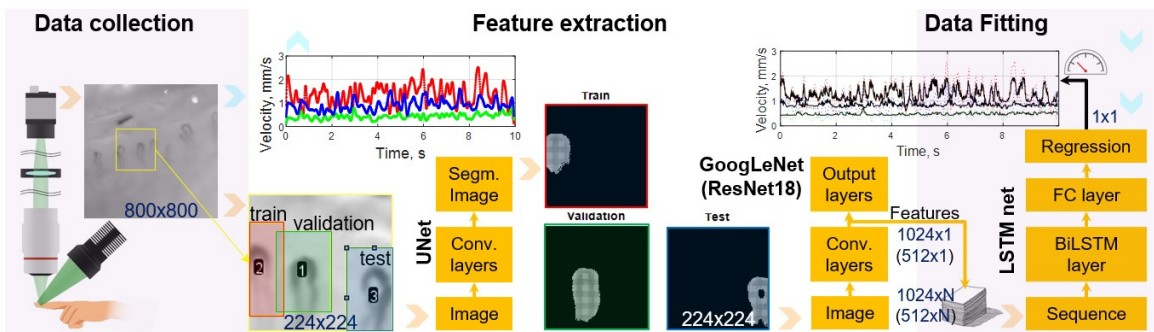

Figure 1: Red blood cells motion detection and velocity approximation procedure includes data collection, preliminary feature extraction, and data fitting

The study involved patients with rheumatic diseases and healthy volunteers presented in more detail in the paper by D. Stavtsev and coauthors  (Stavtsev et al., 2019).

## 3. Data labeling and preliminary feature extraction

Processing of the obtained videos allowed determination the velocity of red blood cells in separate capillaries using the algorithm described in detail in papers by M. Volkov and coauthors  (Gurov et al., 2018). The main idea is that RBCs on a frame have maximal intensities that helps to detect their positions in the video frames and calculate the RBCs velocities. The approach is complex and needs video stabilization. The flow rate in a capillary is periodical with the frequency equal to the heart beating frequency. As the result, each frame corresponds to the velocity value  Figure 1. The next few steps are connected with preliminary feature extraction before the data fitting. The cropped frames include 3 vessels for the training, validation and testing procedures  Figure 1. A part of the dataset of about 300 cropped frames was used as a training set for the semantic segmentation with UNet. The trained UNet provides segmentation for 3 classes for the vessels 1, 2, and 3. The masked frames represent 3 independent subsets for the following feature extraction Figure 1. A deep convolutional network allows to transform an input image to a feature

vector. In this paper the GoogLeNet and the ResNet18 are used Figure 1. The feature vectors are the output of the activations on the last pooling layer (Mat).

## 4. Data fitting using the LSTM network

A sequence of N feature vectors represents a video fragment of the RBCs flow and an input for the LSTM network Figure 1. The network architecture includes a sequence input layer, a BiLSTM layer with NH hidden units, a dropout layer, a fully connected layer and regression layer with one output that is the velocity value (Mat). The test set obtained for the vessel 3 helped to test the trained network with the absolutely new data. The results are presented in Table 4.

Table 1: Accuracy of the mean velocity approximation in the time range of 10 s

| Feature Extractor | Sequence length | LSTM depth | Train | Validation | Test |
|---|---|---|---|---|---|
| GoogLeNet | 15 | 200 | 98.7 | 96.4 | 49.7 |
| GoogLeNet | 15 | 20 | 99.9 | 99.7 | 62.2 |
| ResNet18 | 30 | 20 | 95.4 | 96.0 | 61.3 |
| ResNet18 | *15* | *20* | *96.2* | *92.3* | *96.7* |
| ResNet18 | 7 | 20 | 98.7 | 91.2 | 87.8 |

The first 3 lines in Table 4 demonstrate the overfitting. The best results was obtained with the ResNet18 feature extractor when the sequence length was of 15 and the BiLSTM layer was of 20 hidden units. The accuracy at each time step is small Figure 1. The following research supposed to be connected with application of the bounding boxes detection method to the RBCs motion detection and velocity approximation by tracing their motion. This may help to exclude the velocity data labeling. The more important is that the motion detection and study the RBCs elastic deformation may help to solve problems of health monitoring and diagnostics.

## Acknowledgments

The work is funded by the Russian Science Foundation under the grant No 20-79-00332.

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
