# OpenReview forum: "Application of Deep Supervised Learning to Nailfold videocapillaroscopy and Red Blood Cells Velocity Approximation"
_MIDL.io/2021/Conference/Short — Submitted to MIDL 2021_

### Official Review · Reviewer_D77B · 2021-04-25

**Confidence:** 3
**Final Rating:** 1

**Summary:**

The authors use nested deep neural networks in nailfold high-speed capillaroscopy to detect and quantize red blood cell (RBC) motion. This paper is based on a recently developed technique that the authors describe elsewhere. Using their strategy of nested components they show that they can successfully reach an accuracy of 0.96 for the RBC velocity.

**Strengths:**

Tackling an interesting medical problem to provide real-time information to the examiner. The authors identify correctly weaknesses in their method, i.e. overfitting of the networks. The approach itself is also potentially interesting.

**Weaknesses:**

•	The paper can be largely improved in terms of language and form, e.g. no brackets around Figure references, Table 4 referenced, whereas only Table 1 is available.
•	Emergence of train, validation and test data is unfortunately unclear, 300 frames for training seems rather low in the context of deep neural networks. For example, is this a 2 s recording of a single person? How many have been recorded? Knowing that a short paper has limited space, this information is crucial to interpret sections 3 and 4, and results in Table 1 (4).
•	The feature extractor selection is unclear, GoogLeNet and ResNet-18 seem to be outdated choices in 2021 and no details are given about the exact implementation of the feature extractor.
•	The experiment conditions are not coherent, and based on the low image number predictable. I think any interpretation here is hard, also preliminary in this context of a short conference paper.
•	The data is shared via Google Drive and not a dedicated repository, such as zenodo, that ensures constant access to the community.

**Deanonymize Review:**

no

**Justification Of The Rating:**

I believe that the paper has several strong weaknesses, from language to presentation and novelty of methodology and their implementations. Therefore, I think the paper is of low interest to MIDL in its current form.

**Paper Type:**

both

**Special Issue:**

no

---

### Meta-Review · Area_Chair_3UVy · 2021-05-09

**Recommendation:** Reject
**Confidence:** 5

**Metareview:**

The review highlights numerous problems with the current progress of the work. While MIDL encourages early submission of ongoing research as short paper, this work is not mature enough for publication. I therefore recommend rejection.

---

### Decision · Program_Chairs · 2021-05-11

Reject